# Quantum Stream Cipher Based on Holevo–Yuen Theory

**DOI:** 10.3390/e24050667

**Published:** 2022-05-10

**Authors:** Masaki Sohma, Osamu Hirota

**Affiliations:** Quantum ICT Research Institute, Tamagawa University, Tokyo 194-8610, Japan; sohma@eng.tamagawa.ac.jp

**Keywords:** physical cipher, optical fiber communication, optical satellite communication, quantum communication theory

## Abstract

In this review paper, we first introduce the basic concept of quantum computer-resistant cryptography, which is the cornerstone of security technology for the network of a new era. Then, we will describe the positioning of mathematical cryptography and quantum cryptography, that are currently being researched and developed. Quantum cryptography includes QKD and quantum stream cipher, but we point out that the latter is expected as the core technology of next-generation communication systems. Various ideas have been proposed for QKD quantum cryptography, but most of them use a single-photon or similar signal. Then, although such technologies are applicable to special situations, these methods still have several difficulties to provide functions that surpass conventional technologies for social systems in the real environment. Thus, the quantum stream cipher has come to be expected as one promising countermeasure, which artificially creates quantum properties using special modulation techniques based on the macroscopic coherent state. In addition, it has the possibility to provide superior security performance than one-time pad cipher. Finally, we introduce detailed research activity aimed at putting the quantum stream cipher into practical use in social network technology.

## 1. General View of Cryptography or Cipher in Social Network Systems

At first, we introduce a comment on a general view of cryptography in our research project. In the recent book [1] and a technical paper [2], S. Tsujii, who is one of the leaders of the cyber security community and industry, explains the current situation of the cyber security community and industry on the current trend of the security technology, as follows. “Quantum computer capable of breaking public key cryptographies, such as RSA or elliptic curve cryptography, that relies on mathematical decipherability due to prime number factorization or discrete logarithm problems, will not be developed within 20 years. Nevertheless, the jeopardy due to the cooperative effect with the development of mathematics remains. Thus, NIST is in the process of selecting candidates for quantum computer-resistant cryptography. The applications of cryptography for confidentiality are categorized into the confidential transmission of data itself and the key delivery or storage for that purpose. Then from the viewpoint of academic methods, they are categorized into mathematical cryptography and quantum cryptography. In the former case, there are two types such as public key cryptography and symmetric key cipher. Public key cryptography has the advantage of securely delivering and storing the initial key for data encryption and transmission. However, its processing speed is slow, so symmetric key cipher is responsible for data encryption. On the other hand, quantum cryptography is a cryptographic technique that uses quantum phenomena to improve security performance. The technique that uses quantum communication to perform the key delivery function of public key cryptography is quantum key distribution (QKD: BB-84 et al.), while the technique that uses quantum communication to perform the cryptographic transmission of data itself is called Y-00 quantum stream cipher (see Figure 1). QKD cannot be used to supply keys to One Time Pad cipher, because its data rate is too slow. Y-00 for data encryption is extremely novel in its ability to prevent eavesdroppers from obtaining the ciphertext of the symmetric key cipher. In addition, it is amazing that the strong quantum-ness is created by modulation scheme with multi-ary coherent state signals without any quantum device”.

Let us now turn our focus to quantum cryptography. Both of these quantum technologies are based on designing communication systems to make it difficult for eavesdroppers to steal signals on the communication channels. Such a function to protect the signal itself cannot be realized by mathematical cryptography. As mentioned above, there are two possible system operation methods for these quantum cryptography techniques. One is to use BB-84 quantum key distribution for key delivery and conventional mathematical cryptography for authentication and data encryption. The other is to use Y-00 quantum stream cipher for data encryption and conventional public key cryptography (or quantum computer resistant type) for authentication and key delivery. These quantum cryptography technologies are positioned as technologies to ensure the ultimate security of communication between data center stations, that is of special importance in next-generation 5G and 6G systems. In the following, we will explain the technical contents, applicability to the real world, and development trends.

## 2. Current Status of Quantum Communication Security Technology

### 2.1. Quantum Cryptography

As introduced in the above section, there are two quantum cryptography techniques. Let us give their brief introduction below.

(1)Quantum Key Distribution

BB-84 quantum key distribution (QKD) was proposed by C. H. Bennett and G. Brassard in 1984. It is a protocol to share a secret key sequence by using photon communication, that is guaranteed to be quantum nature. Since the photons used in this protocol are weak light, the transmission speed and distance are limited. In addition, many of the sequence of photons that carry information are lost due to attenuation effects in the transmission line, and the sequence of photons that reaches the receiver is also subject to errors due to noise effects. So, the operation involves discarding the majority of the received bit sequence. Therefore, data itself cannot be sent, only random numbers can be sent. Thus, only the delivery of the secret key for symmetric key cipher is possible. This is why it is called QKD. Recently, many newspapers have reported that several R&D groups can provide the commercial systems of QKD. The transmission speed is the order of 100 Kbit/s, and transmission length is below 100 km. The satellite system is one of the solutions to cope with the distance. However, the transmission speed is so small. In any case, if one tries to increase the transmission speed, then there is a trade-off, and one has to shorten the relay interval. Since the maximum transmission speed is about a megabit, it is difficult to supply keys to the one-time pad cipher for data after key delivery, and it is likely to be limited to supplying initial keys (secret keys) for AES and others.

(2)Quantum Stream Cipher

Y-00 quantum stream cipher is a protocol for physical symmetric key cipher proposed by H.P. Yuen of Northwestern University in the DARPA project (2000) [3]. The details are explained in the next section, but a simple concept is presented here.

This technique is characterized by the fact that it does not allow the physical signals consisting of the mathematical random generator and information data to be obtained without error. In this scheme, the ciphertext in Y-00 circuit system of the mathematical cipher consisting of the generator and data, which is the target of the eavesdropper, as described by $y=\alpha_{i}\left(X,f_{g}\left(K_{s}\right),R_{p}\right)$. Then, we design the system such that the ciphertext $y=\alpha_{i}\left(X,f_{g}\left(K_{s}\right),R_{p}\right)$ is mapped into ensemble of coherent state $|\Psi (X,K_{s},R_{p})>$ with the quantumness based on the Holevo–Yuen theory [4,5,6]. This is called Y-00 signal, which corresponds to ciphertext on the Hilbert space. Thus, the ciphertext as the classical signal is protected by the quantumness. Let us describe it shortly. Although ordinary laser light of high power is used as the transmission signal, signals on the communication channel can be made to have very strong quantum properties in the sense of quantum detection theory [7]. This is the Y-00 principle [3]. That is, a large number of physical binary light communication base is prepared to transmit electric binary data, and the binary data is transmitted by using one communication base which is randomly selected from many communication bases by a mathematical cipher. Let *M* be the number of the base. The optical signals on the communication channel become ultra-multiple-valued signals (2M=4096 or more values are common) against the eavesdropper without the knowledge of communication base. At this time, strong quantum nature in the signal ensemble appears even if the one signal is in high power light, when it is constructed by such ultra-multiple-valued signal. In other words, this method means that the quantum nature in the sense of quantum detection theory [7] is created artificially by modulation schemes, so that it does not require light with strong physical quantum nature, such as a photon. The Y-00 signals of the length *m* (number of slot) are described as follows:(1)$$\begin{array}{ccc} \; & \; & |\Psi \left(X,K_{s},R_{p}\right)>=|\alpha_{i}\left(X,f_{g}\left(K_{s}\right),R_{p}\right){>}_{1}\\ \; & \; & ⊗|\alpha_{j}\left(X,f_{g}\left(K_{s}\right),R_{p}\right){>}_{2}\ldots \ldots \\ \; & \; & ⊗|\alpha_{k}\left(X,f_{g}\left(K_{s}\right),R_{p}\right){>}_{m} \end{array}$$
where $|\alpha_{i}\left(X,f_{g}\left(K_{s}\right),R_{p}\right)>$ is coherent state with amplitude α(·), i,j,k=1,2,3,…2M, *X* is plaintext, $f_{g}\left(K_{s}\right)$ is a mathematical pseudo random function of secret key $K_{s}$, and $R_{p}$ is additional randomization. The set of these coherent states is designed to be strong non-orthogonal property, even if each amplitude of the signals is $|\alpha_{k}\left(X,f_{g}\left(K_{s}\right),R_{p}\right)|\gg 1$.

A legitimate receiver with the knowledge for communication base to which the data is sent can ignore the quantum nature of the data, because it is a binary transmission by high-power signal. That is, one can receive the error-free data. On the other hand, an eavesdropper, who does not know the information of the communication base, must receive a sequence of a ultra-multi-valued optical signal that consists of non-orthogonal quantum states of Equation (Equation 1). The quantum noise generated by quantum measurement based on the Holevo–Yuen theory on quantum detection [8,9,10] masks the received signal, resulting in errors. Thus, even if the eavesdropper tries to record the ciphertext, the masking effect of the quantum noise makes it impossible to accurately recover the ciphertext. This fact is a novel function in the cryptology. Figure 2 shows the scheme of Y-00 principle (Appendix A).

### 2.2. Comparison of Services Based on Each Quantum Cryptosystem

QKD and Y-00 are about 40 and 20 years old, respectively. At the time of their invention, the principle models of both quantum cryptography technologies were not very attractive in terms of security and communication performance. However, nowadays, the systems and security assurance technologies of both technologies have evolved dramatically. Based on the results, business models for security services using these quantum cryptography technologies have been proposed. Figure 3 shows the current status.

## 3. Feature of Quantum Stream Cipher

In the near future, optical networks will move toward even higher speeds, but the Y-00 quantum stream cipher can solve technical requirement from the real world. Since there are few introductions to this technology, we describe the details of this technology in the following section.

### 3.1. Basic Scheme

As explained in the previous section, the quantum stream cipher is expected to accelerate advanced application in future communication systems. The reason for this is that this scheme can utilize ordinary optical communication devices and is compatible with existing communication systems. In its design, optical communication, quantum theory, and cryptography are effectively integrated. Therefore, it is also called “Y-00 optical communication quantum cryptography” in implementation studies. Pioneering research on practical experiment for this system has been reported by Northwestern University [8,9], Tamagawa University [10], and Hitachi Ltd. [11]. Theories of system design for the basic system have been given by Nair and others [12,13,14,15].

Let us explain the principle of Y-00 quantum stream cipher. First, the Y-00 protocol starts by specifying the signal system that use the transmission medium. The actual signal to be transmitted is selected in terms of amplitude or intensity, phase, quadrature amplitude, etc., having coherent state |α〉 in quantum optics. Then, the design is made accordingly. Depending on the type of signal to be used, it is called ISK:Y-00, PSK:Y-00, QAM:Y-00, etc.

Here, one communication base consisting of various binary signals is randomly selected for each data slot. Then, a binary data is transmitted by using the communication base selected. Thus, ultra-multi-valued signals appear to be transmitted on the channel. The eavesdropper has to receive the ultra-multi-valued signal, because they do not know which communication base was selected.

### 3.2. Progress in Security Theory

The BB-84 protocol is a key delivery technique for securely sharing secret key sequences (random numbers). The Y-00 protocol is a symmetric key stream cipher technique for cryptographically transmitting data. As mentioned above, both quantum cryptography techniques enhance security by preventing eavesdroppers from taking the exact signal on the communication channel. The models that explains the principle of such physical technology is called the “basic model”. It is this basic model that can be found in textbooks for beginners.

Let us start with a QKD, such as BB-84. If the basic model of the BB-84 protocol is implemented in a real optical fiber communication system, then it can be eavesdropped. Therefore, in order to guarantee security even in systems with noise and energy loss, a technique that combines error correction and privacy amplification (universal hashing) was proposed, and then a theoretical discussion of security assurance became possible. That is, in 2000, P. Shor, et al. [16] proposed a mathematical security theory for BB-84 on an abstract mathematical model called the Shor model, which was later improved by R. Renner [17]. In brief, the security of the BB-84 protocol is evaluated by quantifying quantum trace distance of the two density operators to the ideal random sequence and the random sequence shared by the real system. This is the current standard theory for the security of QKD. It is very difficult to realize a real system that the quantum trace distance is sufficiently small.

On the other hand, from the beginning, the Y-00 protocol can consider the effects of non-ideal communication systems. As mentioned at the above section, the selection of communication base of the Y-00 protocol is encrypted by conventional mathematical cipher. The Y-00 quantum ciphertext, which is an optical signal, is emitted as the transmission signal. So, the ciphertext of the mathematical symmetric key cipher that an eavesdropper needs to decipher corresponds to the Y-00 quantum ciphertext. However, since the set of ultra-multi-valued signals, which is Y-00 quantum ciphertext, are a non-orthogonal quantum state ensemble, their received signals are inaccurate due to errors caused by quantum noise. Therefore, the discussion based on the computational security of the mathematical cryptographic part of Y-00 mechanism to be attacked is replaced by the problem of combination of information theoretic analysis and computational analysis. However, we should emphasize that the discussion with infinite number or asymptotic theory are not our concern, because our concern is a physical system under practical situation. For example, if an attacker needs circuits of the number of the size of the universe to perform the brute-force attack, the system is unbreakable. Or, if an attacker needs 100 years to collect the ciphertext for trying the cryptoanalysis, it is also impractical and unbreakable.

## 4. Survey of the Mathematical Security Analysis

### 4.1. The Main Story of Security

In the conventional symmetric key cipher, we have
(2)$$H(C|X,f\left(K_{s}\right))=0$$
where *X* is plaintext, $K_{s}$ is secret key, $f(K_{s})$ corresponds to running key and $|f\left(K_{s}\right)\left|\gg \right|K_{s}|$, and *C* is ciphertext. However, in physical cipher system, the eavesdropper cannot do anything without obtaining the ciphertext from the physical signal. In the case of the Y-00 scheme, the eavesdropper has no other way but to observe the non-orthogonal signal, because the Y-00 signals corresponding to the ciphertext in the symmetric key cipher are an ensemble of non-orthogonal quantum states. Thus, the ciphertext that the eavesdropper can obtain are randomized by its quantum nature for any quantum processing by several quantum no-go theorems developed by Holevo and Yuen. This result means that the ciphertext cannot be determined correctly, even if the eavesdropper obtains the secret key $K_{s}$ and the plaintext *X*. That is,
(3)$$H(C|X,f\left(K_{s}\right))\neq 0$$
This is the definition of so called “Random Cipher”. Thus, Y-00 scheme is a typical example of the random cipher. Here, let us describe the security evaluation in the practical setting based on two issues.

(i) The first issue: 

The first issue was raised by the community of cryptology. The question of the cryptocommunity is how to formulate the error or correct estimation of ciphertext based on closeness between the sequence of ciphertext from the Y-00 signals received by the eavesdropper and a true random number sequence. Let us consider a quantum trace distance between density operators on the tensor product Hilbert space that corresponds to the ideal random sequence and the random sequence received by the eavesdropper. It can be denoted by following form, based on the Holevo–Yuen theory on quantum detection:(4)$$\begin{array}{ccc} & & \Delta_{q}=\max_{\Pi }Tr\Pi \left(\sum_{y}p\left(y\right)\rho_{C^{I}C^{E}}^{y}-\rho_{C^{I}}⊗\rho_{C^{E}}\right)\\ & & \Pi :POVM \end{array}$$
In this case, $C^{I}$ is the ideal ciphertext, and $C^{E}$ is the output of the Eve’s receiver. Then, $\rho_{C^{I}}$ corresponds to the density operator for ideal randomness, and that of Eve is $\rho_{C^{E}}$ which depends on the randomization based on quantum noise effect and the artificial scheme designed in the Y-00 scheme.

Closeness of the ciphertext sequence of the eavesdropper to a true random number based on the above equation is evaluated as follows [18]:

**Theorem** **1.**
*Trace distance is bounded by Holevo information, as follows:*

(5)
$$\Delta_{q}^{2}\leq B\chi \left(\epsilon \right)$$

*where B is a constant depending on the definition of relative entropy, and χ(ϵ) is Holevo information from the channel to the eavesdropper.*

(6)
$$\chi \left(\epsilon \right)=S\left(\rho_{C^{E}}\right)-\sum_{y}p\left(y\right)S\left(\rho_{C^{E}}^{y}\right)$$

*where S(ρ) is the von Neumann entropy. The above Holevo information is a decrease function by the appropriate randomization technique under the fixed M.*

*Next, the probability that an eavesdropper can estimate the ciphertext $y=\alpha_{k}\left(X,f_{g}\left(K_{s}\right),R_{p}\right)$ of Y-00 quantum stream cipher is given as follows. Let $\Delta_{q}$ be the trace distance of the quantum density operators between an actual protocol and the ideal one. Then the average guessing probability for ciphertext of Y-00 cipher is bounded as follows:*

(7)
$$\frac{1}{N}\leq P_{guess}\leq \frac{1}{N}+\Delta_{q}\leq \frac{1}{N}+\sqrt{B\chi (\epsilon )}$$

*where $N=2^{|C_{y}|}$. $|C_{y}|$ is the length of binary sequence converted from 2M-ary signal with the length m (number of slot). Thus, the guessing probability for the ciphertext $y=\alpha_{k}\left(X,f_{g}\left(K_{s}\right),R_{p}\right)$ is controlled by Holevo information. In conclusion, under the fixed number of N, one can try to design the randomization technique such that χ(ϵ)→0, and $P_{guess}\to 1/N$. Indeed, the Y-00 scheme provides this situation under ciphertext-only attack.*


(ii) The second issue: 

The next issue is information-theoretic security analysis for symmetric key cipher. In general security analysis for the symmetric key cipher, we have three problems—ciphertext-only attack (COA), statistical attack (SA), and known-plaintext attack (KPA), respectively.

The main issue is that, assess to that information-theoretic security (ITS) can be guaranteed depending on how much ciphertext under COA (or plaintext at KPA) an eavesdropper obtains. Shannon gave the following inequality for general mathematical symmetric key ciphers under ciphertext-only attack:(8)$$H\left(X|C\right)\leq H\left(K_{s}\right)$$
This is called the Shannon limit. Thus, one has the following property under KPA for the conventional additive stream cipher.
(9)$$H(K_{s}|X_{n=|K_{s}|},C_{n=|K_{s}|})=0$$
where $X_{n=|K_{s}|},C_{n=|K_{s}|}$ mean plaintext and ciphertext of the length $n=|K_{s}|$, respectively.

A random physical cipher, such as the Y-00 scheme, may break the above relation. We describe the story of the theory in the following. Here, in the Y-00 scheme, the following is guaranteed:(10)$$H(X|C^{B},f\left(K_{s}\right))=0$$
where $C^{B}$ is the ciphertext received by a legitimate receiver. From here, we discuss the new potential of Y-00 scheme. In the case of a ciphertext-only attack, from Equation (Equation 3), this system provides the ability to break the Shannon limit in the cryptology as follows [19,20]:(11)$$H\left(K_{s}\right)\leq H\left(X_{n}|C_{n}^{E}\right)$$
where $X_{n}$, $C_{n}^{E}$ mean the plaintext sequence and ciphertext sequence of the length *n* received by the eavesdropper, respectively. We emphasize that $C_{n}^{E}$ is different of the original ciphertext created by Y-00 mechanism.

Let us consider statistical attack and the known-plaintext attack. Here, the security evaluation is given by the quantum unicity distance [12,19] under the Holevo–Yuen theory on quantum detection [4,5,6], as follows:(12)$$\begin{array}{ccc} n_{0} & : & \;H(K_{s}|C_{n_{0}}^{E})=0 \end{array}$$(13)$$\begin{array}{ccc} n_{1} & : & \;H(K_{s}|X_{n_{1}},C_{n_{1}}^{E})=0 \end{array}$$
where $n_{0}$ and $n_{1}$ are the unicity distances for ciphertext-only attack and known-plaintext attack, respectively. These mean the number of observations needed to find the secret key with and without known plaintext in the sense of information theoretic security. For exceeded number of $n_{0}$ and $n_{1}$, it still provides the algorithm independent computational security.

The formulae of the unicity distance for the concrete Y-00 scheme were given by Nair et al. [12]. Let us compare Equations (9) and (13). If the Y-00 scheme can provide
(14)$$n_{1}\gg \left|K_{s}\right|,$$
then the Y-00 scheme has the great advantage in comparison with the conventional cipher technology. For more rigorous analysis, we have the following criteria proposed by Yuen.
(15)$$W\left(n\right)=\max_{C^{E}}\max_{K_{s}\in K_{C^{E}}}P\left(K_{s}|C_{n}^{E}\right)$$

Thus, it is possible to evaluate the security of this cipher quantitatively. This is a very significant feature in the history of cryptography.

### 4.2. Randomization Technology

In the early days when Y-00 was invented, the model used was the so-called basic model, and it just explained the principle. In order to achieve sufficient quantitative security, the randomization technique described here is necessary. In the criteria of cryptography by Shannon, such as Equations (12) and (13), the Y-00 scheme has a potential to have excellent quantitative security by additional randomization technology.

In this point of view, we have developed a new concept such as “quantum noise diffusion technology” [13,14]. In addition, several randomizations based on Yuen’s idea [3] have been discussed [21]. Using these techniques, it is expected to have security against known-plaintext attacks on key that cannot be achieved by a conventional cipher, as follows:(16)$$H(X_{n}|C_{n}^{E},K_{s})\neq 0$$
for certain finite $n=n_{2}>\left|K_{s}\right|$ under the condition Equation (Equation 10). This means that one cannot pin-down the data under the finite length of ciphertext with error even if the secret key is provided to the attacker after they have received the Y-00 signals by their instruments [19,20]. This comes from the fact that the ciphertext for attacker is not correct ciphertext. This is called advantage creation based on receivers with key and without key.

This is an amazing capability, and this cannot be achieved even with “One Time Pad Cipher”. However, as the pointed out in the above, these security of abilities are limited to “finite”
$n_{1}$, and $n_{2}$ in principle, and these depend on the randomization technique. The general quantitative evaluation for the concrete randomization is still an open question. In this way, we can say that the Y-00 quantum stream cipher has the ability to provide security that exceeds the performance of conventional cryptography while maintaining the capabilities of ordinary optical communication. To date, there have been several criticisms of the security of the Y-00 principle, but one can see that they all turn out to be based on misunderstandings of the structure and claim of the Y-00 principle.

## 5. Concrete Applications of Quantum Stream Cipher

As mentioned above, the Y-00 quantum stream cipher has not yet reached its ideal performance, but in practical use, it has achieved a high level of security that cannot be achieved with conventional techniques, and it can be said that the ciphers are now at a level where they can be introduced to the market. To date, the development of transceiver for the Y-00 quantum stream cipher has been funded by the university president’s discretionary fund, as well as external funds from the Ministry of Education, Science and Technology (MEXT), and the Defense Acquisition Agency (DEA). Here, we introduce examples of the use case of the Y-00 quantum stream cipher.

### 5.1. Optical Fiber Communication

Large amounts of important data are instantaneously exchanged on the communication lines between data centers where various data are accumulated. It is important from the viewpoint of system protection to eliminate the risk that the data are copied in their entirety from the communication channel. We believe that the Y-00 quantum stream cipher is the best technology for this purpose (see Figure 4). On the other hand, this technology can be used for optical amplifier relay system. Hence, it can apply to the current optical communication systems. Transceivers capable of cryptographic transmission at speeds from one Gbit/s to 10 Gbit/s have already been realized, and by wavelength division multiplexing, a 100 Gbit/s system has been tested. Furthermore, communication distances of 1000 km–10,000 km have been demonstrated. In offline experiments, 10 Tbit/s has been demonstrated. In general, a dedicated line such as dark fiber is required. If we want to apply this technology to network function, then we need the optical switching technology developed by the National Institute of Advanced Industrial Science and Technology (AIST). Thus, in collaboration with AIST and other organizations, we have successfully demonstrated the feasibility of using the Y-00 transceiver in testbed optical switching systems (see Figure 5). Furthermore, Figure 6 shows the recent activities of the experimental research group at Tamagawa University towards practical application to the real world [22,23,24,25,26,27,28,29].

### 5.2. Optical Satellite Communication

The Y-00 quantum stream cipher, which was developed for fiber-optic communications, can also be applied to satellite communications. In satellite communication applications, the rate of operation is an important factor because communication performance depends on the weather conditions. With QKD, it is difficult to keep communications up and running except on clear-air nights. In the case of Y-00, communication by any satellite system can be almost ensured when the weather is clear. In case of bad weather, the effects of atmospheric turbulence and scattering phenomena need to be considered. We are currently analyzing the performance of the system in such cases at 10 Gbps operation [30].

### 5.3. Optical Communication from Base on the Moon to Earth

The Japanese government has initiated a study to increase the user transmission rate of optical space communications from 1.8 Gbps to more than 10 Gbps. Furthermore, in the future, the government aims to achieve higher transmission rates in ultra-long-distance communications required for lunar and planetary exploration. This plan is called LUCAS. We have started to design for an implementation of 1 Gbps communication system at a transmission distance of 380,000 km between the Moon and the Earth using the high-speed performance of the Y-00 quantum stream cipher.

## 6. Future Outlook and Conclusions

The current optical network was not laid out in a planned manner, but was configured by extending the existing communication lines for adapting the demand. In the future, the configuration and specifications of the optical network will be determined following to new urban planning. An actual example is the smart city that Toyota Motor Corporation et al. have disclosed as a future plan. Many ideas are also being discussed in other organizations. Recently, NTT has announced a future network concept so called IOWN. In these systems, the security of the all optical network with ultra-high speed is also important issue. The group of QKD and the group of Y-00 are promoting their respective technologies. However, recently, NSA and others announced the international stance on QKD [31]. They have a negative view of QKD, because the communication performance of QKD based on weak signal is not sufficient for applications to real situations. So, we do not employ QKD for key distribution of the initial key of Y-00, as shown in Figure 3 (Appendix B).

On the other hand, the Y-00 quantum stream cipher is a technology that can realize the specification of high speed and long communication distance. In addition, the signals of Y-00 cipher with ultra-multiple-valued scheme for coherent state signal, so called quantum modulation, can have stronger quantum properties than QKD in the sense of quantum detection theory. So, the security is protected by many quantum no-go theorems (Appendix C). Although it is difficult to make an accurate prediction, there is a good chance that such a new technology will be used in the future. In view of the situation described in this paper, the Y-00 quantum stream cipher will contribute to real-world applications of quantum technology for Society 5.0, and new business development can be expected. Finally, we would like to note that Chinese research institutes have recently been actively working on Y-00 quantum stream cipher. Figure 7 shows a list of academic papers on their activities [32,33,34,35,36,37,38,39]. It is expected that many research institutes will participate in this technological development. 

## Figures and Tables

**Figure 1 entropy-24-00667-f001:**
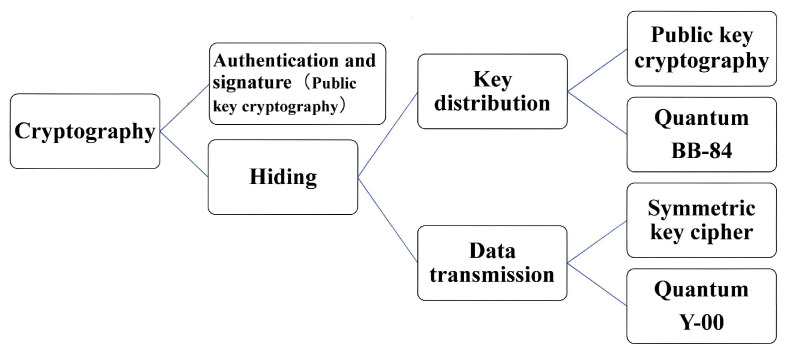
Classification of cryptographic techniques.

**Figure 2 entropy-24-00667-f002:**
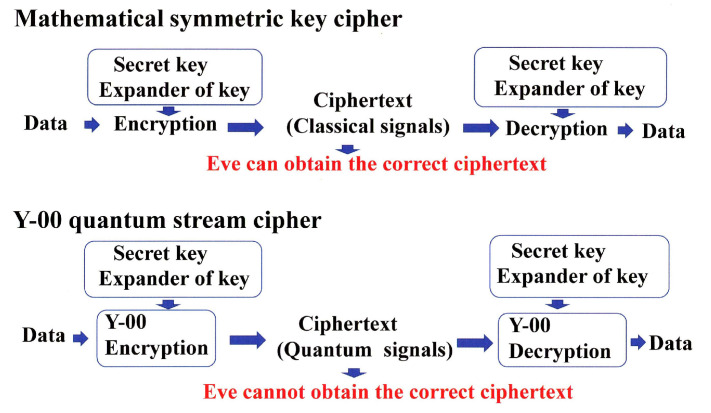
Principle of operation of Y-00 quantum stream cipher. Classical signal means that they have distinguishability, and quantum signal means it is impossible to distinguish them precisely. Y-00 encryption is the function of converting a classical signal into a quantum signal. It is also called quantum modulation.

**Figure 3 entropy-24-00667-f003:**
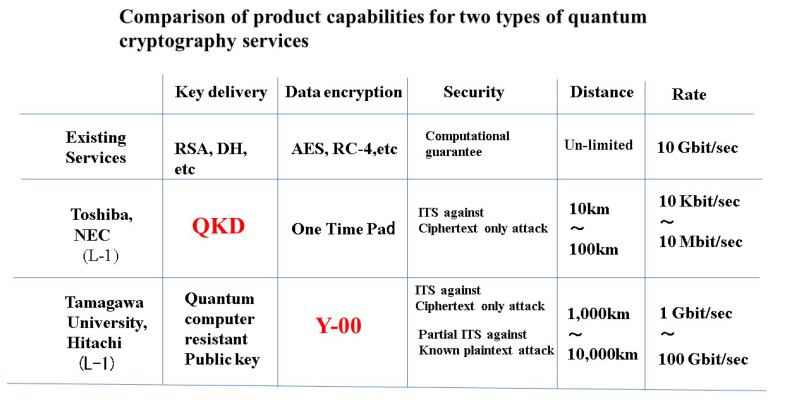
Comparison of product capabilities for two types of quantum cryptography services.

**Figure 4 entropy-24-00667-f004:**
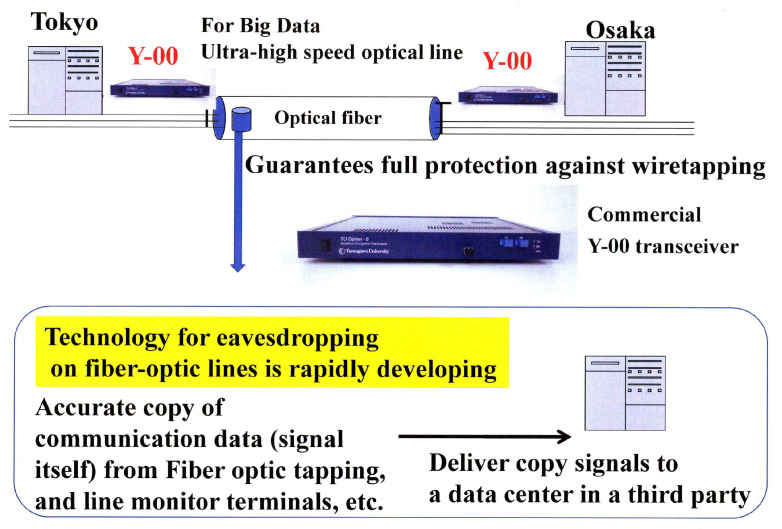
Application to data center communication security (protection against eavesdropping, tampering, and virus injection from communication lines). Commercial transceiver is for 1 Git/s optical ethernet. This can be mass produced.

**Figure 5 entropy-24-00667-f005:**
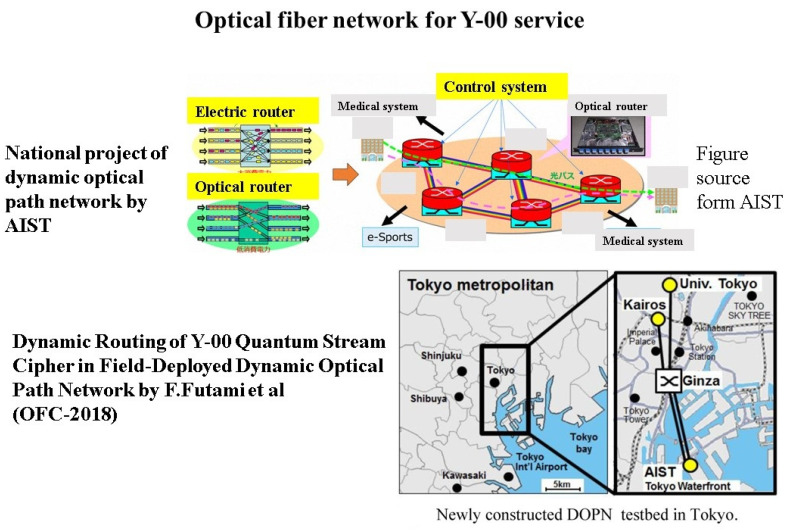
Scheme of optical network by dynamic path and experimental demonstration of service of the Y-00 quantum stream cipher by Tamagawa University and AIST in Tokyo Bay Coastal area.

**Figure 6 entropy-24-00667-f006:**
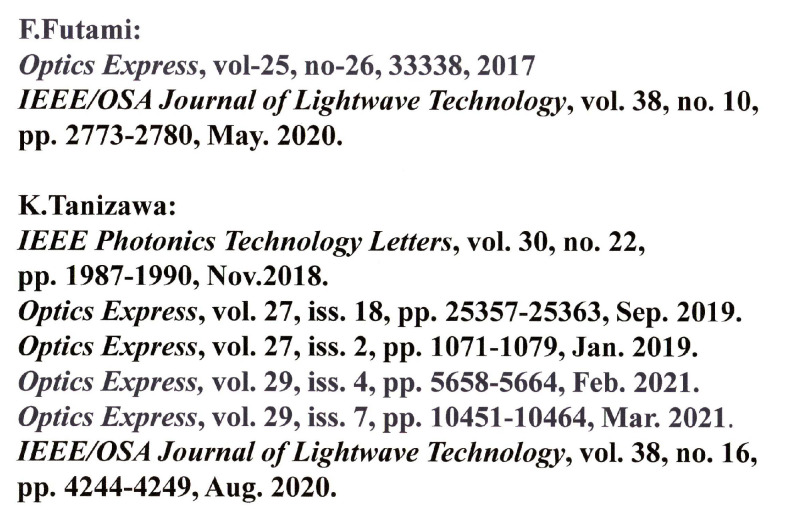
Recent activities of experiment of Y-00 quantum stream cipher at Tamagawa University.

**Figure 7 entropy-24-00667-f007:**
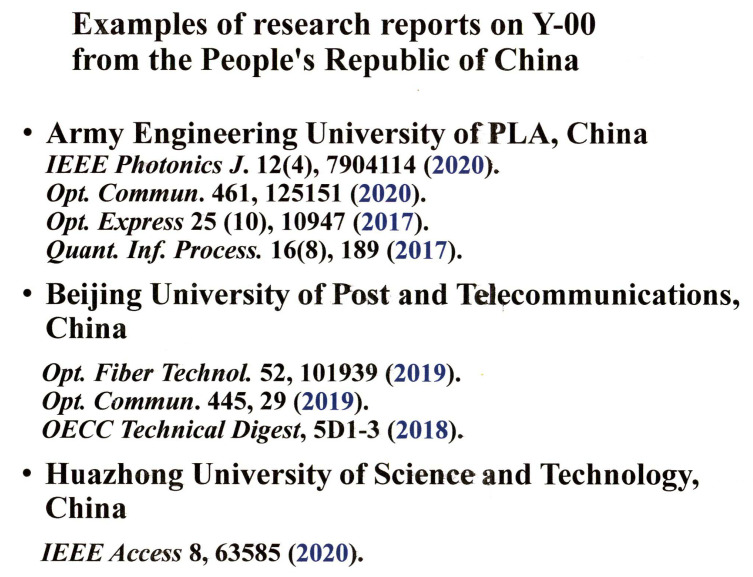
Research activities on the Y-00 quantum stream cipher in China.

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
