# Peer review of "Quantum Stream Cipher Based on Holevo–Yuen Theory"

_entropy, 2022, doi:10.3390/e24050667_

Round 1

Reviewer 1 Report

This paper reviews the state of quantum stream cipher (Y-00) encryption and how it fits into the greater cryptography space. There is probably a place for a paper like this, and these authors are well-suited since they have performed an extensive amount of related work. I am however a bit uncomfortable with some of the wording that may either give a wrong impression or overstate the capability of this type of encryption. I also found the paper hard to follow at points. Some recommendations to remedy these issues are stated below.

General:

  • There is a lot of notation in the paper. Perhaps a chart that lists and defines the terms would be helpful.
  • Two figures just have references in them. I’m not sure how helpful they are. Perhaps instead a chart listing the relevant properties of each reference (e.g. data rate, modulation format, propagation distance…) and the team or country responsible would be more educational.

Abstract:

  • “…it has come to be understood that these methods cannot provide functions that surpass conventional technologies for social systems in the real environment.” I’m not so sure “cannot” would be widely endorsed. It is true there are many restrictions on QKD that will limit its applicability, but I think most would say that under certain conditions it can provide a benefit. I would revise this line accordingly.
  • “the only countermeasure…” I don’t know that “only” is justified. Perhaps it is one promising countermeasure, but have you really shown it is the only one?
  • “…may provide superior security than One Time Pad…” This line must be changed. The OTP may be difficult to apply in many cases, but given that in principle it leaks zero information and Y-00 does leak something the line makes no sense as written.

Section 1.

  • Is there a reference for the S. Tsujii quote?

Section 2.

  • Page 3 line 81-82: “does not allow” is an overly strong statement. The transmitted signal can be intercepted, but noise on the signal means that the underlying mathematical symmetric key cipher cannot be noiselessly reconstructed. I think you need to include something like “noise-lessly” or the like.
  • The above line also highlights a confusion I have when you use the word ciphertext. Normally ciphertext is clearly defined (the encrypted bits), but it can be defined in different ways for quantum stream cipher so I don’t always follow what you mean. In 81-82 I assume it means the exact electrically generated signal that is based on the data and the underlying mathematical symmetric key cipher used, which corresponds to the transmitted signal if it had zero noise. You may also use different types of “ciphertext” in this paper. Can you clearly define them and use a distinctive symbol if there is more than one type of ciphertext? It may help to compare ciphertext from a traditional stream ciphers and a quantum stream cipher early on in the paper to make the point clear.
  • Line 93-94: the quantum nature of relatively strong signals containing many photons per symbol has appreciable noise levels with respect to the separation between the 2M signal levels, but the quantum nature itself has nothing to do with the modulation scheme.
  • Line 97: I think f(K_s) is a mathematical pseudo-random function (not random)
  • Line 107-108: this is similar to the last comment on ciphertext. I again take from this statement that by “accurately” you mean “noiselessly” and by ciphertext you mean the exact classical symbol that would be transmitted in the absence of noise.
  • Would a pictorial depiction of the transmitted state and noise levels help improve Fig. 2 (or put in another figure)? A picture may be helpful to the reader.

Section 3:

  • Line 177: should be “in-practice unbreakable”

Section 4:

  • Line 185: typo on “question”
  • Equation (5); did define all these subscripts/superscripts?
  • Equation (6); did you ever define S ?
  • Words between (6) and (7); the eavesdropper can estimate the ciphertext right away, just with noise. I guess you are stating the probability Eve can exactly guess the ciphertext.
  • Lines 195-197. While perhaps true, these lines don’t have much meaning without knowing how fast N and Holovo information go to zero. In practice I assume the Holovo information will approach zero much slower than 1/N, so P_guess doesn’t really go to 1/N. Can you comment on how small the guessing probability would be- perhaps using an example? Note here you use |C| for “binary ciphertext”. Can you define what this means?
  • Iine 205-207 is helpful, but this can also be stated earlier so the paper is more easily followed
  • What is n_0 and n_1? I see a reference but can you just define it here for us? I guess it is the number of observations needed to find the key, with and without the plaintext, but you should write it down.
  • Equ (16) would indeed be amazing, if true. I don’t believe it though, as if so then Bob could not recover the data. I tried to find this equation in [18] but couldn’t. There is of course no way it can be more secure than OTP so that doesn’t really make sense to me. This section 4.1 is very short and makes some very strong claims. I think it needs to be re-written, clarifying (moderating) your statements and assumptions. (16) especially should be explained or better yet dropped entirely.

Section 5:

  • 226-227: why have Y-00 not reached ideal performance and what would be needed to reach it? Can you comment on the security provided in current practice?
  • Line 286: “stronger quantum properties than QKD”- I’m not sure where you get this from. It works with larger signal levels, which is good in practice but I wouldn’t say it is stronger.

Reviewer 2 Report

The authors present a brief but self-contained overview of the advantages and benefits of the quantum stream cipher protocol in secure quantum communications. Specifically, they focus on the Y-00, which ensures superior security functions (longer communication distances, higher transmission rates, higher levels of security) over other conventional single-photon-based encryption techniques. This overview not only offers a comparison between the Y-00 protocol and the usual BB-84+QKD, but it also provides the interested reader with both a simple but enlightening formal (mathematical) discussion of the protocol, followed by what seems to be feasible applications and reach in real world, which might also feed the interest of the quantum communications community.

In my opinion, this work is timely and, despite of its brevity, provides the reader with a wide perspective on a topic of strategical interest at present, with an also interesting view on how the scientific part permeates the social one through different funded proposals. Therefore, I find it suitable for publication in its present form.

Author Response

We would like express our sincere appreciation to the reviewer for his accurate understanding of the purpose of this paper and his explanation of the content of the paper.

Reviewer 3 Report

The manuscript is of the review character. It describes mathematical and quantum cryptography concepts. The Authors start their presentation form introducing some basic concepts necessary to understand further considerations presented in the article. Then, they discuss the current state of the art in quantum communication security technologies concentrating on the quantum key distribution and quantum stream cipher. The Authors give a short characteristic of exemplary already existing solutions. Next, applying mathematical security analysis, they discuss the Y-00 scheme and concentrate on the possible issues and potential attacks on the systems and, finally, randomization technology. At the end of the paper, some applications of quantum stream ciphering are presented, where optical fiber and satellite communication methods are mentioned and discussed in the context of their future applications.

The manuscript is well written in general and is accessible to Readers who are not necessarily experts in the field. Moreover, the Authors provide necessary references helpful in further studies. Although the article is a short review, it could be stimulating for future research. Finally, I can conclude that the manuscript can be accepted for publication.

Author Response

We would like to express our sincere appreciation to the reviewer for his accurate understanding of the purpose of this paper.